# Level Compressibility of Certain Random Unitary Matrices

**DOI:** 10.3390/e24060795

**Published:** 2022-06-07

**Authors:** Eugene Bogomolny

**Affiliations:** Université Paris-Saclay, CNRS, LPTMS, 91405 Orsay, France; eugene.bogomolny@lptms.u-psud.fr

**Keywords:** level compressibility, barrier billiards

## Abstract

The value of spectral form factor at the origin, called level compressibility, is an important characteristic of random spectra. The paper is devoted to analytical calculations of this quantity for different random unitary matrices describing models with intermediate spectral statistics. The computations are based on the approach developed by G. Tanner for chaotic systems. The main ingredient of the method is the determination of eigenvalues of a transition matrix whose matrix elements equal the squared moduli of matrix elements of the initial unitary matrix. The principal result of the paper is the proof that the level compressibility of random unitary matrices derived from the exact quantisation of barrier billiards and consequently of barrier billiards themselves is equal to 1/2 irrespective of the height and the position of the barrier.

## 1. Introduction

The leading idea behind statistical descriptions of complex deterministic quantum problems is that quantum characteristics (e.g., eigenenergies) of a large variety of such problems are so erratic and irregular that their precise values are irrelevant (such as the position of a molecule in the air), and only their statistical properties are of importance. As matrices are inherent in quantum mechanics, random matrices occupy a predominant place in the application of statistics to quantum problems [1]. In a typical setting, one tries to find a random matrix ensemble whose eigenvalues have the same statistical distributions as (high-excited) eigenenergies of a given deterministic quantum problem. Until now, this query has been figured out only for two limiting classes of quantum problems: (i) models whose classical limit is integrable [2] and (ii) models whose classical limit is chaotic [3]. For generic integrable models, quantum eigenenergies are distributed as eigenvalues of diagonal matrices with independent identically distributed (i.i.d.) elements which means that their correlation functions after unfolding coincide with the ones of the Poisson distribution [2]. For generic chaotic systems, it was conjectured in [3] that their eigenenergies are distributed as eigenvalues of standard random matrix ensembles (GOE, GUE, GSE) depending only on system symmetries whose correlation functions are known explicitly [4]. The difference between these two cases is clearly seen from the limiting behaviour of their nearest-neighbour distribution $P_{0}\left(s\right)$, which is the probability density that two eigenvalues are separated by a distance *s* and there are no other eigenvalues in between. For the Poisson statistics, there is no level repulsion that means that $\lim_{s\to 0}P_{0}\left(s\right)\neq 0$ and for large argument $P_{0}\left(s\right)$ decreases exponentially with *s*. Standard random matrix ensembles levels repel each other, $\lim_{s\to 0}P_{0}\left(s\right)=0$, and $P_{0}\left(s\right)∼\exp\left(-as^{2}\right)$ when s→∞.

These two big conjectures form a cornerstone of quantum chaos and have been successfully applied to various problems from nuclear physics to number theory. Nevertheless, they do not cover all possible types of models. Especially intriguing is the class of pseudo-integrable billiards (see, e.g., [5]) which are two-dimensional polygonal billiards whose angles $\theta_{j}$ are rational multiples of π
$$\theta_{j}=\frac{m_{j}}{n_{j}}\pi$$
with co-prime integers $m_{j}$ and $n_{j}$. A peculiarity of such billiards is seen in the fact that their classical trajectories belong to a two-dimensional surface of genus *g* related with angles as follows [6]
$$g=1+\frac{N_{n}}{2}\sum_{j}\frac{m_{j}-1}{n_{j}}$$
where $N_{n}$ is the least common multiple of all denominators $n_{j}$. Consequently, any such model with at least one numerator $m_{j}>1$ is neither integrable (which would imply that trajectories belong to a two-dimensional torus with g=1) nor fully chaotic (in which case trajectories should cover a three-dimensional surface of constant energy), and the aforementioned conjectures cannot be applied to such systems. Numerical calculations show that for many pseudo-integrable billiards, the spectral statistical properties of corresponding quantum problems differ from both the Poisson statistics and the random matrix statistics mentioned above (see, e.g., [7,8] and references therein). In particular, for these models, (i) $\lim_{s\to 0}P_{0}\left(s\right)=0$ as for standard random matrix ensembles but (ii) $P_{0}\left(s\right)∼\exp\left(-bs\right)$ for large *s* as for the Poisson statistics. Such hybrid statistics, labeled intermediate statistics, had been first observed in the Anderson model at the metal–insulate transition [9,10], and they constitute a special, interesting but poorly investigated class of spectral statistics.

Probably the simplest example of pseudo-integrable systems is the so-called barrier billiard, which is a rectangular billiard with a barrier inside sketched in Figure 1a. The quantum problem for this model consists of solving the Helmholtz equation
$$\left(\Delta +E_{\alpha }\right)\Psi_{\alpha }\left(x,y\right)=0$$
imposing that eigenfunction $\Psi_{\alpha }\left(x,y\right)$ obeys (e.g.,) the Dirichlet conditions on the boundary of the rectangle as well as on the barrier
$$\Psi_{\alpha }\left(x,y\right){{|}_{\mathrm{boundary}}=0,\;\Psi_{\alpha }\left(x,y\right)|}_{\mathrm{barrier}}=0.$$

Calculating the exact *S*-matrix for the scattering inside the infinite slab with a barrier depicted in Figure 1b, it has been demonstrated in [7,8] that spectral statistics of this model is the same as the statistics of eigenvalues of the following N×N random unitary matrix
(1)$$B_{m,n}=e^{i\Phi_{m}}\frac{L_{m}L_{n}}{x_{m}+x_{n}},\;m,n=1,\ldots ,N$$
where $\Phi_{m}$ are i.i.d. random variables uniformly distributed on interval [0,2π) and
(2)$$L_{m}=\sqrt{2x_{m}\prod_{k\neq m}\frac{x_{m}+x_{k}}{x_{m}-x_{k}}}$$
where coordinates $x_{m}$ depend on the position of the barrier.

Define the following quantities (momenta) of propagating modes in each of three channels indicated in Figure 1b.
$$p_{m}^{\left(1\right)}=\sqrt{k^{2}-\frac{\pi^{2}m^{2}}{b^{2}}}\;,\;p_{m}^{\left(2\right)}=\sqrt{k^{2}-\frac{\pi^{2}m^{2}}{{\left(b-h\right)}^{2}}}\;,\;p_{m}^{\left(3\right)}=\sqrt{k^{2}-\frac{\pi^{2}m^{2}}{h^{2}}}\;.$$

If the ratio h/b is an irrational number, coordinates $x_{m}$ have the following form
(3)$$\vec{x}=b\left(\underset{N_{1}}{\underset{︸}{p_{1}^{\left(1\right)},\ldots ,p_{N_{1}}^{\left(1\right)}}},\underset{N_{2}}{\underset{︸}{-p_{1}^{\left(2\right)},\ldots ,-p_{N_{2}}^{\left(2\right)}}},\underset{N_{3}}{\underset{︸}{-p_{1}^{\left(3\right)},\ldots ,-p_{N_{3}}^{\left(3\right)}}}\right).$$
Here, $N_{j}$ with j=1,2,3 are the numbers of propagating modes in each channel
(4)$$N_{1}=\left[\frac{kb}{\pi }\right],\;N_{2}=\left[\frac{k(b-h)}{\pi }\right],\;N_{3}=\left[\frac{kh}{\pi }\right]$$
where [x] is the largest integer ≤x and the total dimension of the *B*-matrix is $N=N_{1}+N_{2}+N_{3}$.

When the ratio h/b is a rational number, h/b=p/q with co-prime integers *p* and *q* (p<q), there exist exact plane wave solutions of barrier billiard equal to zero at the whole line passing through the barrier. It is natural to disregard them and take into account only non-trivial eigenvalues. In such case, coordinates $x_{m}$ have to be chosen as indicated below
(5)$$\vec{x}=b\left(\underset{k\neq 0\;\mathrm{mod}\;q}{\underset{︸}{p_{1}^{\left(1\right)},\ldots ,p_{k}^{\left(1\right)},\ldots ,p_{N_{1}}^{\left(1\right)}}},\underset{N_{2}}{\underset{︸}{-p_{1}^{\left(2\right)},\ldots ,-p_{N_{2}}^{\left(2\right)}}},\underset{k\neq 0\;\mathrm{mod}\;p}{\underset{︸}{-p_{1}^{\left(3\right)},\ldots ,-p_{k}^{\left(3\right)},\ldots ,-p_{N_{3}}^{\left(3\right)}}}\right)\;.$$
The dimension of this vector is
(6)$$N^{\left(r\right)}=N_{1}+N_{2}+N_{3}-2N_{0},\;N_{0}=\left[\frac{kb}{\pi q}\right].$$

The matrix *B* can be generalised for arbitrary vector $x_{m}$ provided the following interlacing conditions are fulfilled
$$|x_{1}\left|>\right|x_{2}\left|>\ldots >\right|x_{N}|,\;x_{m}={\left(-1\right)}^{m+1}\left|x_{m}\right|.$$

Exact correlation functions for the *B*-matrix are unknown at present. In [7,8], it was argued that an approximate Wigner-type surmise for this matrix corresponds to the so-called semi-Poisson distribution [11]. In particular, it implies that the probability density $P_{n}\left(s\right)$ that two levels are separated by a distance *s* and there are exactly *n* levels in between (after the standard unfolding) is given by the following expression
$$P_{n}\left(s\right)=\frac{2^{2n+2}}{(2n+1)!}s^{2n+1}e^{-2s},\;n\geq 0\;.$$
Numerical calculations presented in [7,8] agree with these simple formulas.

Despite the simplicity of the semi-Poisson distribution, it has been observed (mostly numerically) in various models ranging from certain pseudo-integrable models and quantum maps (see references in [7,8]) to the entanglement spectra of two-bits random many body quantum circuits [12].

This paper is devoted to the calculation of another important characteristic of spectral statistics, namely the level (or spectral) compressibility, which is a long-range characteristic of the spectral two-point correlation function. In particular, this quantity determines the limiting behaviour of the variance of the number of levels inside a given interval. More precisely, let N(L) be the number of eigenvalues in an interval *L* unfolded to the unit mean density, which means that the mean number of levels in interval *L* equals *L*, 〈N(L)〉=L. By definition, the number variance is $\Sigma^{\left(2\right)}\left(L\right)\equiv \langle {\left(N\left(L\right)-L\right)}^{2}\rangle$. If for large *L*
(7)$$\Sigma^{\left(2\right)}\left(L\right)\underset{L\to \infty }{⟶}\chi L$$
constant χ is called the level compressibility. The importance of this quantity follows from the fact that for integrable systems with the Poisson statistics, χ=1, but for standard random matrix statistics typical for chaotic models, χ=0. For all examples of intermediate statistics, it was observed that 0<χ<1.

The conventional way of determining the level compressibility for dynamical systems is the summation over all periodic orbits in the diagonal approximation initiated in [13]. For the symmetric barrier billiard with h/b=1/2 and d/a=1/2, it has been demonstrated in [14] that χ=1/2. The same value had been obtained in [15] for the case h/b=1/2 and arbitrary barrier height d/a. Finally, for h/b=p/q with co-prime integers *p*, *q* and irrational values of d/a, it has been proven in [16] that
$$\chi =\frac{1}{2}+\frac{1}{q}$$
but in the calculations, exact eigenvalues whose eigenfunctions are zero on the whole line y=h (see Figure 1a) have not been excluded. When these trivial eigenvalues are removed, the answer is χ=1/2 [17]. Therefore, direct (and quite tedious) calculations suggest that for (almost) all positions and heights of the barrier, the level compressibility is the same as for the semi-Poisson statistics [11]
(8)$$\chi =\frac{1}{2}$$
but the reason of this universality remains obscure.

The purpose of the paper is to find analytically the spectral compressibility for barrier billiards and for a few other models directly from the corresponding random unitary matrices. To achieve the goal, it is convenient to slightly generalise the method developed in [18] for random unitary matrices that appeared in the quantisation of quantum graphs [19]. The method is briefly explained in Section 2. In Section 3, this method is applied to random unitary matrices derived in [20] by the quantisation of a simple interval-exchange map. In this case, the transition matrix is a circulant matrix whose eigenvalues are known explicitly. The results coincide with the exact level compressibility for these models obtained in [21,22,23,24]. This example gives credit to the method and permits explaining its main features without unnecessary complications. In particular, it clarifies the situation (not covered by Refs. [21,22,23,24]) when a parameter that entered the matrix takes an irrational value. Numerically, it has been observed [22] that in such case, the spectral statistics are well described by the ones of chaotic systems (GOE or GUE), although the Lyapunov exponent of the underlying classical map is always zero.

The main part of the paper is devoted to the derivation of the level compressibility for random matrices associated with barrier billiards (Equation 1). The calculations are more complicated, as no eigenvalues (except one) are known analytically. The simplest case of the symmetric barrier billiard with h/b=1/2 is investigated in Section 4. To obtain tractable expressions, a kind of paraxial approximation is developed, which permits controlling the largest terms. By using such approximation, the transition matrix is transformed into a Toeplitz matrix with a quickly decreasing symbol, which allows us to find its eigenvalues for large matrix dimension. The result of this section is that the level compressibility equals 1/2 in accordance with periodic orbit calculations in [14,15].

In Section 5, random unitary matrices corresponding to barrier billiards with irrational ratio h/b (with coordinates given by (Equation 3)) are considered. In this case, the transition matrix in paraxial approximation contains quickly oscillating terms and, consequently, has forbidden zones in the spectrum. In spite of that, one can argue that largest moduli eigenvalues are insensitive to fast oscillations and are determined solely by a matrix averaged over such oscillations. The final matrix is also a Toeplitz type, which allows analytical calculations, proving that the level compressibility is again equal to 1/2.

Section 6 addresses the calculation of level compressibility in the most complicated case of barrier billiards with rational ratio h/b=p/q≠1/2. The computation is cumbersome, but in the end, one comes to the conclusion that the level compressibility remains equal 1/2. In other words, the level compressibility of barrier billiards is universal (i.e., independent of the barrier position and its height) and coincides with the semi-Poisson prediction [11].

Section 7 gives a brief summary of the results. A few technical points are discussed in Appendix A, Appendix B and Appendix C.

## 2. Generalities

It is well known that the level compressibility (Equation 7) is related with the two-point correlation form factor K(τ) as follows
(9)$$\chi =\lim_{\tau \to 0}K\left(\tau \right)\;.$$

For N×N random unitary matrices *U*, the form factor can, conveniently, be written in the following concise form (see, e.g., [18])
(10)$$K\left(\tau \right)=\langle \frac{1}{N}{\left|\mathrm{Tr}\;U^{n}\right|}^{2}\rangle \;,\;\tau =\frac{n}{N}$$
where the average is taken either on different realisations of random parameters, or over a small window of τ, or both.

Unitary matrices considered in the paper all have the product form
(11)$$U_{j,k}=e^{i\Phi_{j}}w_{j,k}$$
where $\Phi_{j}$ are i.i.d. random phases uniformly distributed between 0 and 2π, and matrix $w_{j,k}$ is a fixed unitary matrix.

In [18] (see [19] for more detailed discussion), it was shown that for such unitary matrices, the averaging over random phases leads in the diagonal approximation to the following formula
(12)$$K^{\left(\mathrm{diag}\right)}\left(\frac{n}{N}\right)=g\frac{n}{N}\mathrm{Tr}\;\left(T^{n}\right)$$
where matrix elements of matrix *T*, called below the transition matrix, are squared moduli of matrix elements of matrix *U*
(13)$$T_{j,k}=|U_{j,k}{|}^{2}={\left|w_{j,k}\right|}^{2}\;.$$
For systems without time-reversal invariance, g=1, and for models with time-reversal invariance, g=2 (here, only cases of g=1 are considered). Due to the unitarily of matrix *U*, the *T*-matrix is a double stochastic matrix, $\sum_{j}T_{j,k}=1$ and $\sum_{k}T_{j,k}=1$, thus having the meaning of classical transition matrix.

Let $\Lambda_{\beta }$ be eigenvalues of the *T*-matrix
$$T_{j,k}u_{k}\left(\beta \right)=\Lambda_{\beta }u_{j}\left(\beta \right)\;,\;\beta =0,\ldots ,N-1\;.$$
From (Equation 12), it follows that
(14)$$K^{\left(\mathrm{diag}\right)}\left(\frac{n}{N}\right)=\frac{n}{N}\sum_{\beta =0}^{N-1}\Lambda_{\beta }^{n}\;.$$

The unitarity imposes that one eigenvalue $\Lambda_{0}=1$. Perron–Frobenius theorem states that all other eigenvalues $|\Lambda_{\beta }|\leq 1$.

Let the set of eigenvalues be ordered $1=\Lambda_{0}\geq \Lambda_{1}\geq \ldots \geq \Lambda_{N-1}$. One has
(15)$$K^{\left(\mathrm{diag}\right)}\left(\tau \right)=\tau \left(\mathrm{Tr}\;\left(T^{n}\right)\right)=\tau +\tau \sum_{\beta =1}^{N-1}\Lambda_{\beta }^{N\tau },\;\sum_{\beta =1}^{N-1}\Lambda_{\beta }^{N\tau }\leq \left(N-1\right){\left|\Lambda_{1}\right|}^{N\tau }\;.$$

It has been noted in [18] that if
(16)$$\lim_{N\to \infty }{\left|\Lambda_{1}\right|}^{N}=0$$
then the second term in (Equation 15) goes to zero for all finite τ and $K^{\left(\mathrm{diag}\right)}\left(\tau \right)=\tau$ for small τ. Consequently, it is reasonable to conjecture (as it has been done in [18]) that the whole spectral statistics of such matrices will be well described by standard random matrix formulas.

For the matrices discussed below, criterium (Equation 16) is not fulfilled. Instead, in all considered cases, the largest moduli eigenvalues of transition matrices have the form 1−O(1/N). To calculate the level compressibility from (Equation 14), the summation over all such eigenvalues is performed analytically, and then, the limit τ→0 is taken.

It is well known that the form factor is not a self-average quantity. It has strong fluctuations and necessarily requires a smoothing. There exist two different sources of fluctuations. The first is related with random phases in matrices (Equation 11). Equation (Equation 12) corresponds to the averaging over these random phases in the diagonal approximation. The second has its roots in non-smoothness of $K^{\left(\mathrm{diag}\right)}\left(n/N\right)$ for different *n* and could be removed by a smoothing over a small interval of τ. (A trivial example is ${\left(-1\right)}^{n}$).

## 3. Interval-Exchange Matrices

This section is devoted to the calculation of the level compressibility for special unitary matrices derived in [20] by quantisation of a simple two-dimensional parabolic map. Slightly generalising their result [21], one can write these matrices in the following form
(17)$$M_{n,m}=e^{i\Phi_{n}}\frac{1-e^{2\pi i\alpha N}}{N\left(1-e^{2\pi i(n-m+\alpha N)/N}\right)}\;,\;n,m=1,\ldots ,N\;.$$
Here, α is a real parameter and $\Phi_{n}$ are i.i.d. random variables uniformly distributed between 0 and 2π (the case with ‘time-reversal symmetry’ when $\Phi_{N-n+1}=\Phi_{n}$ requires only multiplication of the formulas below by factor g=2, as indicated in (Equation 12)).

When α is a rational number α=p/q with co-prime integers *p* and *q*, the original classical map is an interval-exchange map and, as it was shown in detail in [21,22,23,24], spectral statistics of matrices (Equation 17) in the limit of N→∞ are unusual and peculiar. It appears that the limiting results depend on the residue of pNmodq (when pN≡0modq matrix (Equation 17) have explicit eigenvalues not interesting for our purposes). For example, if α=1/5, there are two possibilities: N≡±1 mod 5 and N≡±2mod5. In the first case, the nearest-neighbour distribution is
$$P_{0}\left(s\right)=\frac{5^{5}}{4!}s^{4}e^{-5s}$$
but for the second one, the exact result is different
$$P_{0}\left(s\right)=\left(a_{2}s^{2}+a_{3}s^{3}+a_{4}s^{4}+a_{5}s^{5}+a_{6}s^{6}\right)e^{-5s}$$
where $a_{2}=625/2-275\sqrt{5}/2$, $a_{3}=3125/2-1375\sqrt{5}/2$, $a_{4}=71875/48+33125\sqrt{5}/48$, $a_{5}=-15625/3+9375\sqrt{5}/4$, and $a_{6}=1015625/288-453125\sqrt{5}/288$.

Although the spectral correlation functions for α=p/q are different for different residues pNmodq, the calculations show that the spectral compressibility for all residues remains the same [21,22]
(18)$$\chi =\frac{1}{q}\;.$$

Matrices (Equation 17) with irrational α were investigated numerically in [22], and it was observed that their spectral statistics are well described by standard random matrix ensembles (GOE or GUE). In particular, it implies that in such case
(19)χ=0.
Below, it is shown that values (Equation 18) and (Equation 19) can easily be recovered by the discussed method.

The transition matrix (Equation 13) for the discussed case has the form
(20)$$T_{n,m}\equiv {\left|M_{n,m}\right|}^{2}=\frac{\sin^{2}\left(\pi \alpha N\right)}{N^{2}\sin^{2}\left(\pi (n-m+\alpha N)/N\right)}\;.$$
This is a circulant matrix, and its eigenvalues are simply the Fourier transforms of its matrix elements
(21)$$\Lambda_{\beta }=\frac{\sin^{2}\left(\pi \alpha N\right)}{N^{2}}\sum_{s=0}^{N-1}\frac{e^{-2\pi i\beta s/N}}{\sin^{2}\left(\pi (s+\alpha N)/N\right)}\;.$$

Differentiating both sides of the identity with integer *s* on *z*
$$\sum_{m=0}^{N-1}e^{2\pi im(s+z)/N}=\frac{1-e^{2\pi iz}}{1-e^{2\pi i(s+z)/N}}\;,$$
after straightforward transformations, one proves that
$$\frac{\sin^{2}\left(\pi z\right)}{\sin^{2}\left(\pi (s+z)/N\right)}=-2i\sin\left(\pi z\right)e^{-\pi iz}\sum_{m=0}^{N-1}\left(m-N\frac{e^{\pi iz}}{2i\sin(\pi z)}\right)e^{2\pi im(s+z)/N}$$
from which it follows that eigenvalues (Equation 21) are
(22)$$\Lambda_{\beta }=\left(1-\frac{\beta }{N}\left(1-e^{-2\pi i\alpha N}\right)\right)e^{2\pi i\beta \alpha }\;,\;\beta =0,1\;\ldots ,N-1\;.$$
Notice that $\Lambda_{0}=1$, $\Lambda_{N-\beta }=\Lambda_{\beta }^{*}$ for β=1,…,N−1.

Consider first the case of rational α=p/q. Assume that N≠0 mod *q* and calculate the form factor from (Equation 14) separately for n≡r mod *q* with r=0,1,…,q−1 (i.e., n=qt+r with integer *t*)
Kqt+rN=qt+rN1+2Re∑β=1N/21−βN1−e−2πiαNqt+re2πiβpr/q.
Here, we do not take into account that when *N* is even the term with β=N/2 is real and has an additional factor 1/2.

Notice that the phase depends only on *r*. For large *N*, one can put the first factor in the exponent and sum the geometric progression from 1 to infinity. The answer is
(23)Kr(τ)=τ+2τRee−ξτ+2πipr/q1−e−ξτ+2πipr/q,τ=qt+rN
where $\xi =1-e^{-2\pi ipN/q}=1-e^{-2\pi ik/q}$ with k=pNmodq.

When τ→0, all terms except the one with r=0 tend to zero, as they do not have a pole at τ=0. The remaining term equals
limτ→0K0(τ)=2Re1ξ=2Re(1−e2πiz)2−2cos(2πz)=1.
It means that after the averaging over random phases, the limiting value of the form factor $K_{r}\left(\tau \right)$ strongly depends on small changes of τ=(qt+r)/N. For r=0, $K_{0}\left(\tau \right)$ tends to 1 for small τ≪1 but for all other terms with r=1,…,q−1$K_{r}\left(\tau \right)$ tends to 0. The difference between these different values of τ is very small, of the order of 1/N. Therefore, after the averaging over any small (but finite) interval of τ, one gets
$$\lim_{\tau \to 0}\langle K\left(\tau \right)\rangle =\frac{1}{q}$$
which agrees with (Equation 18) obtained in [22] by a different method.

For illustration, the results of the direct calculation of the form factor for α=1/5 and N=399 and N=398 are presented in Figure 2. First, eigenvalues of the matrix (Equation 17) were calculated numerically, and then using (Equation 10), the form factor for different *n* values was computed. The result is averaged over 1000 realisations of random phases. It is clear that, indeed, for different residues of *n* modulo 5, the results are different, and when n≡0 mod 5, the form factor at small argument is close to 1, but for all other residues, it starts at 0. The average over all 5 residues begins at 1/5, as expected.

Such a clear picture appears when the form factor is calculated at special values of τ, τ=n/N with integer *n*. Computing it at arbitrary arguments leads to an irregular plot but, of course, the average curve remains unchanged.

Exactly the same formulas can be applied for an irrational value of parameter α. In this case, one has
(24)K(τ)=τ+2τRee−ζτ1−e−ζτ,ζ=1−e−2πiαN−2πiαN.

The exponent $\zeta =2\sin^{2}\left(\pi \alpha N\right)+i\left[\sin(2\pi \alpha N)-2\pi \alpha N\right]$ has a large imaginary part when N→∞. It means that the above expression is a strongly oscillated function of τ. When averaged over a small interval of τ, one obtains K(τ)=τ as it should be for the ensemble of usual random matrices (GUE). This result follows without calculations from the fact that the average of all eigenvalues $\Lambda_{\beta }^{\tau N}$ except β=0 equals zero as a consequence of rapidly changing phases. (For even *N*, the term with β=N/2 is real, but as it tends to zero at large *N*, its contribution is negligible).

Notice that criterion (Equation 16) for matrix (Equation 17) with irrational α is not fulfilled. Nevertheless, the spectral statistics of such a matrix is close to GUE statistics. This example illustrates a new mechanism for the appearance of random matrix statistics. The contribution of higher eigenvalues of the transition matrix (Equation 13) decreases not because of a gap between the first and the second eigenvalues as has been proposed in (Equation 16) but due to rapid oscillations for large matrix dimensions.

## 4. Symmetric Barrier Billiard with *h*/*b* = 1/2

The central problem of the paper is the determination of level compressibility for the *B*-matrices given by (Equation 1) and (Equation 2) by employing the method proposed in [18] and used in the previous section for matrices derived from the quantisation of an interval-exchange map. The simplicity of treatment of interval-exchange matrices comes from the fact that their transition matrices are circulant matrices whose eigenvalues are known exactly. For the *B*-matrices, calculations are more complicated, as there are no explicit formulas for eigenvalues of the corresponding transition matrix.
(25)$$T_{m,n}={\left|B_{m,n}\right|}^{2}=\frac{L_{m}^{2}\;L_{n}^{2}}{{\left(x_{m}+x_{n}\right)}^{2}},\;m,n=1,\ldots ,N\;.$$

This section is devoted to the investigation of the *B*-matrix corresponding to the symmetric barrier billiard with ratio h/b=1/2. In this case, q=2, b−h=h, and the second part of the vector $\vec{x}$ in (Equation 5) coincides with the third one. Now, trivial eigenfunctions can be removed by considering a desymmetrised rectangular billiard with height h=b/2 and imposing the Neumann boundary conditions for negative *x* and y=h. It is the equivalent of dropping the second part of vector (Equation 5) and taking coordinates $x_{m}$ as follows [7]
(26)$$x_{m}={\left(-1\right)}^{m+1}b\sqrt{k^{2}-\frac{\pi^{2}m^{2}}{b^{2}}}\;,\;m=1,\ldots ,N\;,\;N=\left[\frac{kb}{\pi }\right]\;.$$
Odd (resp., even) indices describe the first (resp., the third) part of vector (Equation 5).

The numerically calculated spectrum of the transition matrix in this case is presented in Figure 3a.

To calculate this spectrum (or, at least, the behaviour of largest-moduli eigenvalues) analytically, a kind of paraxial approximation has been developed. It is based on the fact that the main ingredient of matrices with intermediate statistics is a linear fall-off of matrix elements from the diagonal [25,26]. In the simplest setting, it means that
$$M_{m,n}∼\frac{R_{m,n}}{m-n+\mathrm{const}.}\;,\;m,n\gg 1\;,\;m-n=O\left(1\right)\;.$$

Therefore, it is natural to assume that the most important contributions come from the pole terms with $R_{m,n}\approx R_{m,m}$. This type of approximation can be done directly from the definition (Equation 2), as it is demonstrated in Appendix A. According to these results, the *T*-matrix in the paraxial approximation is a block matrix
(27)$$T=\left(\begin{array}{cc} 0_{o,o} & t_{o,e}\\ t_{o,e}^{T} & 0_{e,e} \end{array}\right)\;,\;t_{o,e}\equiv t_{2m-1,2n}=\frac{1}{\pi^{2}{\left(n-m+1/2\right)}^{2}}\;.$$
Here, subscripts ‘o’ and ‘e’ indicate odd and even indices, respectively.

It is instructive to obtain this answer without the knowledge of the exact *B*-matrix. One can achieve it by using the instantaneous approximation used in quantum mechanics when the interaction changes suddenly. In optics, such an approximation is analogous to the Fraunhofer diffraction. In the barrier billiard, it corresponds to the situation when a wave with large momentum quickly moving in a channel enters into another channel (cf., Figure 1b). In the instantaneous approximation, eigenfunctions in the new channel are just a re-expansion of the initial eigenfunctions into a complete set of eigenfunctions with correct boundary conditions inside the final channel.

Consider a normalised wave with the Neumann boundary conditions at y=h=b/2 and the Dirichlet ones at y=0
$$\psi_{2m-1}^{\left(1\right)}\left(x,y\right)=\frac{2}{\sqrt{b}}\sin\left(\frac{(2m-1)\pi }{b}y\right)\exp\left(ip_{2m-1}^{\left(1\right)}x\right)\;,\;x<0,\;0\leq y\leq b/2$$
propagating in the desymmetrised barrier billiard at negative *x*. When it penetrates into the region of positive *x*, it has to be expanded into correct waves propagating inside that region
$$\psi_{2m-1}^{\left(1\right)}\left(x,y\right)=\sum_{n=1}^{\infty }S_{2m-1,2n}\psi_{2n}^{\left(3\right)}\left(x,y\right)$$
where $\psi_{2n}^{\left(3\right)}\left(x,y\right)$ are waves obeying the Dirichlet boundary conditions at y=0 and y=h=b/2
$$\psi_{2n}^{\left(3\right)}\left(x,y\right)=\frac{2}{\sqrt{b}}\sin\left(\frac{2\pi n}{b}y\right)\exp\left(ip_{n}^{\left(3\right)}x\right)\;,\;x>0,\;0\leq y\leq b/2\;.$$

Coefficients $S_{2m-1,2n}$ are the *S*-matrix for this process. In the paraxial approximation, they are calculated as follows (notice that in the paraxial approximation, $p_{2m-1}^{\left(1\right)}\approx p_{n}^{\left(3\right)}$)
$$\begin{array}{ccc} S_{2m-1,2n}^{\left(p.a.\right)} & = & \frac{4}{b}\int_{0}^{b/2}\sin\left(\frac{\pi (2m-1)}{b}\right)\sin\left(\frac{2\pi n}{b}y\right)dy=\\ & = & \frac{{\left(-1\right)}^{m+n}}{\pi^{2}\left(n-m+1/2\right)}+\frac{{\left(-1\right)}^{m+n}}{\pi^{2}\left(n+m-1/2\right)}\;. \end{array}$$
Taking into account only the pole term (and symmetry of the *S*-matrix), one obtains for the *T*-matrix exactly the same expression as (A1).

Thus, the transition matrix (Equation 27) is a block Toeplitz matrix. It is plain that its eigenvalues $\Lambda =\pm \sqrt{\lambda_{\beta }}$ where $\lambda_{\beta }$ are eigenvalues of a $N_{1}\times N_{1}$ matrix (with $N_{1}=\left[N/2\right]$).
$$P_{m,n}=\sum_{k=1}^{N_{1}}t_{m,k}t_{n,k},\;t_{m,k}=\frac{1}{\pi^{2}{\left(k-m+1/2\right)}^{2}}\;.$$

Dominant contributions to the sum come from regions k∼m and k∼n. Due to a quick decrease of the summands, the finite summation over *k* can safely be substituted in the limit $N_{1}\to \infty$ by the sum over all integer *k*
$$P_{m,n}\approx \sum_{k=-\infty }^{\infty }t_{m,k}t_{n,k}\;.$$
Using (Equation 54), the necessary sum is easily calculated, and the result is
(28)$$P_{m,n}=t_{m-n}\;,\;t_{0}=\frac{1}{3}\;,\;t_{r\neq 0}=\frac{2}{\pi^{2}r^{2}}\;.$$
This formula is valid when m,n≫1 and m−n=O(1).

Matrix (Equation 28) is a Toeplitz matrix with quickly decreasing matrix elements. It is well known that eigenvalues of the N×N Toeplitz matrix can be asymptotically calculated as follows (see, e.g., [27,28,29,30,31] and references therein)
(29)$$\lambda_{\beta }=f\left(\frac{\beta }{2N}+O\left(\frac{1}{N}\right)\right)\;,\;\beta =1,\ldots ,N$$
where function f(x) called the symbol is the Fourier series of $t_{r}$.
$$f\left(x\right)=\sum_{r=-\infty }^{\infty }t_{r}e^{2\pi irx}\;.$$
(More precise formulas can be found in the above references).

Using (Equation 52) and (Equation 53) one finds that the symbol of matrix (Equation 28) is
$$f\left(x\right)=\frac{1}{3}+\frac{2}{\pi^{2}}\overset{\infty }{\underset{r=-\infty }{\sum^{\prime }}}\frac{e^{2\pi irx}}{r^{2}}=\frac{1}{3}+4B_{2}\left(x\right)={\left(1-2x\right)}^{2}\;.$$

Therefore, eigenvalues of the *P*-matrix for large $N_{1}$ are
$$\lambda_{\beta }\approx {\left(1-\frac{\beta }{N_{1}}\right)}^{2},\;\beta =1,\ldots ,N_{1}\;.$$

Eigenvalues of block matrix (Equation 27) $\Lambda =\pm \sqrt{\lambda_{\beta }}$. Taking into account that the dimension of matrix (Equation 27) is $N\approx 2N_{1}$, one concludes that approximately its eigenvalues are
(30)$$\Lambda =\pm \left(1-\frac{2\beta }{N}\right)\;,\;\beta =1,\ldots ,\frac{1}{2}N\;.$$
With the corresponding redefinition of index β, these eigenvalues can be rewritten in the form
$$\Lambda_{\beta }\approx \frac{2\beta }{N}-1,\;\beta =1,\ldots ,N$$
which agrees well with numerical calculations (see Figure 3a).

The form factor in the diagonal approximation is related with transition matrix eigenvalues by (Equation 14)
$$K^{\left(\mathrm{diag}\right)}\left(\frac{n}{N}\right)=\frac{n}{N}\sum_{\beta }\Lambda_{\beta }^{n}\;.$$
As $\Lambda_{N-\beta }=-\Lambda_{\beta }$ for β=1,…,N−1 (which is a consequence of the block structure of the transition matrix (Equation 27)), the form factor $K^{\mathrm{diag}}\left(n/N\right)$ with odd *n* in the diagonal approximation tends to zero when τ=n/N→0
(31)$$K^{\left(\mathrm{diag}\right)}\left(\tau \right)\underset{\tau \to 0}{⟶}0\;.$$
However, for even *n*, one obtains a different answer. Equation (Equation 30) may not be accurate for extreme eigenvalues with small β. For τ=2n/N, one can separate the contribution of small $\beta <\beta_{0}$ and the rest for which (Equation 30) is a good approximation
(32)$$K^{\left(\mathrm{diag}\right)}\left(\tau \right)=\tau \left(\mathrm{const}+2\sum_{\beta =\beta_{0}}^{N/2}{\left(1-\frac{2\beta }{N}\right)}^{\tau N}\right)\underset{N\to \infty }{⟶}\tau \left(\mathrm{const}+\frac{2e^{-2j_{0}\tau }}{1-e^{-2\tau }}\right)\underset{\tau \to 0}{⟶}1\;.$$
As has been discussed in the previous section, it means that the spectral compressibility of the *B*-matrix for symmetric barrier billiard coincides with the semi-Poisson value
(33)$$\chi \equiv K\left(0\right)=\frac{1}{2}\;.$$

For illustration, the form factor for the symmetric billiard calculated numerically by the direct diagonalisation of 400×400 matrices (Equation 1) with coordinates given by (Equation 26) and averaged over 1000 realisations is shown in Figure 3b. Two branches corresponding to odd and even *n* are clearly seen. The average over odd and even values agrees well with the semi-Poisson expression for the form factor [11] and, in fortiori, the level compressibility is 1/2, as in (Equation 33).

## 5. Barrier Billiard with Irrational Ratio *h*/*b*

The transition matrices for a general barrier billiard with an off-centre barrier remain the same as in (Equation 25), but coordinates $x_{m}$ should have the form (Equation 3) for irrational ratio h/b and (Equation 5) for rational h/b=m/q. The direct calculations of eigenvalues of these matrices reveal that they are more complicated that the ones for symmetric billiard with h/b=1/2 discussed in the previous section. As an example, in Figure 4, the spectra of the transition matrices with $h/b=1/\sqrt{5}$ and h/b=2/5 are presented. It is clearly seen that although eigenvalues with small moduli are quite irregular and have gaps, the largest moduli eigenvalues are well described by a straight line $\Lambda_{\beta }=2\beta /N-1$.

This section is concentrated on the analytical treatment of billiards with irrational ratio h/b. As in the previous section, the first step consists of the calculation of a paraxial *S*-matrix for the scattering inside the slab with a barrier, as shown in Figure 1b. It can easily be completed in the instantaneous approximation exactly as above. In such an approximation, only transitions from channel 1 to 2 and to 3 and their inverse are non-zero. One has
$$\begin{array}{ccc} S_{m,n}^{1\to 3} & = & \frac{2}{\sqrt{bh}}\int_{0}^{h}\sin\left(\frac{\pi m}{b}y\right)\sin\left(\frac{\pi n}{h}y\right)dy=\\ & = & \frac{{\left(-1\right)}^{n}\sin\left(\pi hm/b\right)}{\pi \sqrt{bh}}\left({\left(m/b-n/h\right)}^{-1}-{\left(m/b+n/h\right)}^{-1}\right)\;. \end{array}$$
Similarly,
$$\begin{array}{ccc} S_{m,n}^{1\to 2} & = & \frac{2}{\sqrt{b(b-h)}}\int_{h}^{b}\sin\left(\frac{\pi m}{b}y\right)\sin\left(\frac{\pi n}{b-h}\left(b-y\right)\right)dy=\\ & = & \frac{{\left(-1\right)}^{n}\sin\left(\pi mh/b\right)}{\pi \sqrt{b(b-h)}}\left({\left(m/b-n/\left(b-h\right)\right)}^{-1}-{\left(m/b+n/\left(b-h\right)\right)}^{-1}\right)\;. \end{array}$$

The transition matrix $T={\left|S\right|}^{2}$ also has the same block structure. Retaining only the pole (the first) terms (and slightly changing the notations), one obtains
(34)$$T=\left(\begin{array}{ccc} 0_{1,1} & t_{n_{1},n_{2}}^{1\to 2} & t_{n_{1},n_{3}}^{1\to 3}\\ t_{n_{1},n_{2}}^{1\to 2} & 0_{2,2} & 0_{2,3}\\ t_{n_{1},n_{3}}^{1\to 3} & 0_{3,2} & 0_{3,3} \end{array}\right)$$
where $n_{i}=1,\ldots ,N_{i}$ with $N_{i}$ given by (Equation 4) and
(35)$$t_{n_{1},n_{2}}^{1\to 2}=\frac{z_{2}\sin^{2}\left(\pi n_{1}h/b\right)}{\pi^{2}{\left(z_{2}n_{1}-n_{2}\right)}^{2}}\;,\;t_{n_{1},n_{3}}^{1\to 3}=\frac{z\sin^{2}\left(\pi n_{1}h/b\right)}{\pi^{2}{\left(zn_{1}-n_{3}\right)}^{2}}\;,\;z_{2}=1-z,\;z=\frac{h}{b}\;.$$

Due to the block structure of the transition matrix (Equation 34), it follows that its eigenvalues Λ are determined by the relation $\Lambda^{2}=\lambda_{\beta }$ where $\lambda_{\beta }$ are eigenvalues of $N_{1}\times N_{1}$ matrix
$$P_{m,n}=\sum_{k=1}^{N_{2}}t_{m,k}^{1\to 2}t_{n,k}^{1\to 2}+\sum_{k=1}^{N_{3}}t_{m,k}^{1\to 3}t_{n,k}^{1\to 3}\;.$$
For large matrix dimensions, the summation can be extended over all integer *k*, and the sums can be calculated explicitly by using (Equation 55) from Appendix C. The results are
(36)$$P_{m,m}=\frac{z^{2}+{\left(1-z\right)}^{2}}{3}\left(3-2\sin^{2}\left(\pi mz\right)\right)\;,$$
and for m≠n
(37)$$P_{m,n}=\frac{2\left(\sin^{2}\left(\pi zm\right)+\sin^{2}\left(\pi nz\right)\right)}{\pi^{2}{\left(m-n\right)}^{2}}-\frac{2(1-2z)\sin(\pi mz)\sin(\pi nz)\sin(\pi (m-n)z)}{\pi^{3}z\left(1-z\right){\left(m-n\right)}^{3}}\;.$$
This matrix is a combination of Toeplitz terms dependent on the difference m−n and oscillating terms (which explains the existence of forbidden zones in its spectrum; see Figure 4a).

Due to the unitarity of the *B*-matrix, the exact transition matrix $T={\left|B\right|}^{2}$ has the largest eigenvalue equal to 1, whose corresponding eigenvector is (1,1,…,1). It is natural (and is confirmed by calculations) that eigenvectors of the *P*-matrix corresponding to large moduli eigenvalues are slowly varying functions. Consequently, all oscillating terms in (Equation 36) and (Equation 37) for large *m* and *n* could be ignored. These arguments lead to the following recipe of the next step of approximation. Put m=n+r and average all matrix elements of the *P*-matrix over quickly changing phase πnz. The calculations are straightforward and
(38)$$\langle P_{m,n}\rangle \equiv \lim_{N\to \infty }\frac{1}{N}\sum_{n=1}^{N}P_{n+r,n}=f_{m-n}$$
where
$$f_{0}=\frac{2(z^{2}+{\left(1-z\right)}^{2})}{3}\;,\;f_{r\neq 0}=\frac{2}{\pi^{2}r^{2}}-\frac{\left(1-2z)\sin(2\pi rz\right)}{2\pi^{3}z\left(1-z\right)r^{3}}\;.$$
Eigenvalues of such a matrix for large *N* are calculated by the Fourier transform of this symbol
$$\lambda_{\beta }=\sum_{r=-\infty }^{\infty }f_{r}e^{2\pi irx},\;x=\frac{\beta }{2N_{1}}\;.$$

The necessary sums are expressed through the Bernoulli polynomials (Equation 52), (Equation 53), and the result is
$$\lambda_{\beta }=\frac{2}{3}\left(z^{2}+{\left(1-z\right)}^{2}\right)+4B_{2}\left(x\right)-\frac{1-2z}{3z(1-z)}\left(B_{3}\left(\left\{x+z\right\}\right)-B_{3}\left(\left\{x-z\right\}\right)\right)\;.$$

From the beginning, one can assume that h<h−b, i.e., z=h/b<1/2 (the case h/b=1/2 was discussed in Section 4). Then
(39)$$\lambda_{\beta }=\left\{\begin{array}{cc} 1-4x+\frac{x^{2}}{z(1-z)}\;, & 0\leq x\leq z\\ \frac{1-2z+2z^{2}}{1-z}-\frac{2x(1-x)}{1-z}\;, & z\leq x\leq \frac{1}{2} \end{array}\right.,\;x=\frac{\beta }{2N_{1}}\;.$$
As eigenvalues of the block matrix (Equation 34) $\Lambda =\pm \sqrt{\lambda_{\beta }}$, it follows that close to maximum value (i.e., with small β),
$$\Lambda \approx \pm \left(1-\frac{\beta }{N_{1}}+O\left(\frac{1}{N_{1}}\right)\right)\approx \pm \left(1-\frac{2\beta }{N}\right)\;.$$

As in the calculation of the form factor (Equation 14) small moduli eigenvalues are irrelevant, one can ignore higher-order terms in the above expression, which gives the same expression as in (Equation 30). It means that the level compressibility of barrier billiards with irrational ratio h/b has the same value as in the preceding sections $\chi =\frac{1}{2}$.

In Figure 5a, the above formulas are compared with the results of direct calculations for the *P*-matrix with $h/b=1/\sqrt{5}$. As has been demonstrated, the approximate expression (Equation 39) is tangent to the exact spectrum close to 1. The form factor computed numerically for the same ratio h/b is presented in Figure 5b. The agreement with the above result is clearly seen.

## 6. Barrier Billiard with Rational Ratio *h*/*b* = *p*/*q*


The calculation of transition matrix eigenvalues when the ratio h/b is a rational number can be completed by a similar method. An additional difficulty in such case is that one has to select special combinations of states in the second and the third channels to remove trivial eigenvalues equal to zero on the whole line passing through the barrier. It has been discussed in detail in [8] and briefly reviewed in Appendix B. Combining all terms together, one concludes that the transition matrix when h/b=p/q with *p* and *q* being co-prime integers has the block form similar to (Equation 34) but with one more block
(40)$$T=\left(\begin{array}{cccc} 0_{1,1} & t_{n_{1},n_{2}}^{1\to 2} & t_{n_{1},n_{3}}^{1\to 3} & t_{n_{1},n4}^{1\to 4}\\ t_{n_{1},n_{2}}^{1\to 2} & 0_{2,2} & 0_{2,3} & 0_{2,4}\\ t_{n_{1},n_{3}}^{1\to 3} & 0_{3,2} & 0_{3,3} & 0_{3,4}\\ t_{n_{1},n_{4}}^{1\to 4} & 0_{4,2} & 0_{4,3} & 0_{4,4} \end{array}\right).$$
Here, indices $n_{j}=1,\ldots N_{j}^{\left(r\right)}$ have the following restrictions
$$n_{1}≢0\;\mathrm{mod}\;q\;,\;n_{2}≢0\;\mathrm{mod}\;q-p\;,\;n_{3}≢0\;\mathrm{mod}\;p$$
and
(41)$$N_{1}^{\left(r\right)}=N_{1}-N_{0}\;,\;N_{2}^{\left(r\right)}=N_{2}-N_{0}\;,\;N_{3}^{\left(r\right)}=N_{3}-N_{0}\;,\;N_{4}^{\left(r\right)}=N_{0}$$
with $N_{1},N_{2},N_{3}$ given by (Equation 4) and $N_{0}$ is determined by (Equation 50) or (Equation 6). The total matrix dimension is $N^{\left(r\right)}=\sum_{j=1}^{4}N_{j}^{\left(r\right)}=N_{1}+N_{2}+N_{3}-2N_{0}$ as in (Equation 6).

Matrices $t^{1\to 2}$ and $t^{1\to 3}$ are the same as in (Equation 35) and $t^{1\to 4}$ given by (Equation 51) from Appendix B
$$t_{n_{1},n_{4}}^{1\to 4}=\frac{\sin^{2}\left(\pi mh/b\right)}{\pi^{2}p\left(q-p\right){\left(n_{1}/q-n_{4}\right)}^{2}}\;.$$

The eigenvalues of block matrix (Equation 40) Λ are $\Lambda =\lambda_{\beta }^{2}$ where $\lambda_{\beta }$ with $\beta =1,\ldots ,N_{1}$ are eigenvalues of matrix (superscript (res), indicating that the matrix describes the resonance case h/b=p/q)
$$P_{m,n}^{\left(\mathrm{res}\right)}=\sum_{k\neq 0\;\mathrm{mod}\;(q-p)}t_{m,k}^{1\to 2}t_{n,k}^{1\to 2}+\sum_{k\neq 0\;\mathrm{mod}\;p}t_{m,k}^{1\to 3}t_{n,k}^{1\to 3}+\sum_{k}t_{m,k}^{1\to 4}t_{n,k}^{1\to 4}\;.$$
Using an evident relation
$$\sum_{k\neq 0\;\mathrm{mod}\;r}f\left(k\right)=\sum_{k}f\left(k\right)-\sum_{k}f\left(rk\right)$$
and (Equation 55), the above sums can be explicitly calculated.

The results are
(42)$$\begin{array}{ccc} P_{m,m}^{\left(\mathrm{res}\right)} & = & \frac{1}{3q^{2}}\left(p^{2}+{\left(q-p\right)}^{2}\right)\left(3-2\sin^{2}\left(\pi pm/q\right)\right)\\ & + & \frac{2\sin^{4}\left(\pi pm/q\right)\left(3-2\sin^{2}\left(\pi m/q\right)\right)}{3p\left(q-p\right)q^{2}\sin^{4}\left(\pi m/q\right)} \end{array}$$
and when m≠n
(43)$$P_{m,n}^{\left(\mathrm{res}\right)}=\frac{b_{m,n}}{{\left(m-n\right)}^{2}}-\frac{c_{m,n}\sin\left(\pi p\left(m-n\right)/q\right)}{{\left(m-n\right)}^{3}}-\frac{d_{m,n}\sin\left(\pi \left(m-n\right)/q\right)}{{\left(m-n\right)}^{3}}$$
where
(44)$$\begin{array}{ccc} b_{m,n} & = & \frac{2}{\pi^{2}}\left(\sin^{2}\left(\pi pm/q\right)+\sin^{2}\left(\pi pn/q\right)\right)\\ & + & \frac{2\sin^{2}\left(\pi pm/q\right)\sin^{2}\left(\pi pn/q\right)}{p\left(q-p\right)\pi^{2}}\left(\frac{1}{\sin^{2}\left(\pi m/q\right)}+\frac{1}{\sin^{2}\left(\pi n/q\right)}\right)\;,\\ c_{m,n} & = & \frac{2q(q-2p)}{\pi^{3}p\left(q-p\right)}\sin\left(\pi pm/q\right)\sin\left(\pi pn/q\right)\;,\\ d_{m,n} & = & \frac{4q\sin^{2}\left(\pi pm/q\right)\sin^{2}\left(\pi pn/q\right)}{\pi^{3}p\left(q-p\right)\sin\left(\pi m/q\right)\sin\left(\pi n/q\right)}\;. \end{array}$$

Although these expressions are indexed by integers *m* and *n*, this notation is symbolic. The point is that by construction, these integers cannot be arbitrary but have to be not divisible by *q*. Let us order such numbers and let ν(k) with k=1,2,…, be the $k^{\mathrm{th}}$ integer ≢0modq. Then, indices of matrix $P_{m,n}^{\left(\mathrm{res}\right)}$ have to be considered as follows: m=ν(j), n=ν(k) with $j,k=1,2,\ldots ,N_{1}^{\left(r\right)}$ with $N_{1}^{\left(r\right)}$ defined in (Equation 41). In such notation, matrix $P^{\left(\mathrm{res}\right)}$ is $N_{1}^{\left(r\right)}\times N_{1}^{\left(r\right)}$ matrix
$$P_{m,n}^{\left(\mathrm{res}\right)}\equiv P_{\nu (j),\nu (k)}^{\left(\mathrm{res}\right)},\;j,k=1,\ldots ,N_{1}^{\left(r\right)}.$$

The next step, as in the previous section (cf., (Equation 38)), consists in the substitution instead of the above exact expressions, their mean values with fixed difference between the indices
$$\langle P_{m,n}^{\left(\mathrm{res}\right)}\rangle \left(r\right)=\lim_{N\to \infty }\frac{1}{N}\sum_{n=1}^{N}P_{n+r,n}^{\left(\mathrm{res}\right)}$$
where both integers *n* and n+r have to be not divisible by *q*.

According to (Equation 42) and (Equation 43), the $P^{\left(\mathrm{res}\right)}$ matrix is a mixture of functions depending explicitly on the differences of indices and certain coefficients depending on indices modulus *q*. Only the latter requires the explicit averaging. Using (Equation 56)–(Equation 59) from Appendix C, one obtains that
$$\begin{array}{ccc} & & \sum_{m=1}^{q-1}P_{m,m}=\frac{(p^{2}+{\left(q-p\right)}^{2})\left(2q-3\right)}{3q^{2}}+\frac{2p^{2}\left(2q-3p\right)}{3\left(q-p\right)q^{2}},\\ & & \overset{q-1}{\underset{m=1}{\sum^{\prime }}}b_{m,m+r}=\frac{2q(q-p+1-2\sin^{2}\left(\pi pr/q\right))}{\pi^{2}\left(q-p\right)},\\ & & \overset{q-1}{\underset{m=1}{\sum^{\prime }}}c_{m,m+r}=\frac{q^{2}\left(q-2p\right)\cos\left(\pi pr/q\right)}{\pi^{3}p\left(q-p\right)},\;\overset{q-1}{\underset{m=1}{\sum^{\prime }}}d_{m,m+r}=\frac{q^{2}\sin\left(2\pi pr/q\right)}{\pi^{3}p\left(q-p\right)\sin\left(\pi r/q\right)}\;. \end{array}$$
Here, it is taken into account that p/q<1/2. The superscript ′ in these sums indicates that the term with m+r≡0modq is omitted. The latter condition implies that the number of independent terms equal q−1 if r≡0modq or q−2 otherwise. Finally, one obtains
(45)$$\langle P_{m,n}^{\left(\mathrm{res}\right)}\rangle =\alpha_{1}\delta_{m,n}+f_{m-n}$$
with
$$f_{m-n}=\left\{\begin{array}{cc} \frac{\alpha_{2}}{{\left(m-n\right)}^{2}}, & (m-n)\equiv 0\;\mathrm{mod}\;q\\ \frac{\alpha_{3}}{{\left(m-n\right)}^{2}}+\frac{\alpha_{4}\cos\left(2\pi pr/q\right)}{{\left(m-n\right)}^{2}}+\frac{\alpha_{5}\sin\left(2\pi pr/q\right)}{{\left(m-n\right)}^{3}}, & (m-n)\equiv r\neq 0\;\mathrm{mod}\;q \end{array}\right.$$
where constants $\alpha_{j}$ are
$$\begin{array}{ccc} \alpha_{1} & = & \frac{(p^{2}+{\left(q-p\right)}^{2})\left(2q-3\right)}{3q^{2}\left(q-1\right)}+\frac{2p^{2}\left(2q-3p\right)}{3\left(q-p\right)q^{2}\left(q-1\right)}\;\alpha_{2}=\frac{2q(q-p+1)}{\pi^{2}\left(q-p\right)\left(q-1\right)},\\ \alpha_{3} & = & \frac{2q}{\pi^{2}\left(q-2\right)},\;\alpha_{4}=\frac{2q}{\pi^{2}\left(q-p\right)\left(q-2\right)},\;\alpha_{5}=\frac{q^{2}\left(2p-q-2\right)}{2\pi^{3}p\left(q-p\right)\left(q-2\right))}\;. \end{array}$$

Although this matrix depends only on the difference of indices m−n, it is not a Toeplitz matrix as *m* and *n* are not arbitrary numbers but only integers not divisible by *q*. Nevertheless, one can argue that the largest eigenvalues for large matrix dimension can be calculated by a formula similar to Toeplitz matrices (which is a kind of variational method)
(46)$$\lambda_{\beta }=\frac{1}{N_{1}^{\left(r\right)}}\sum_{j,k=1}^{N_{1}^{\left(r\right)}}\langle P_{\nu (j),\nu (k)}^{\left(\mathrm{res}\right)}\rangle e^{2\pi ix(j-k)},\;x=\frac{\beta }{2N_{1}^{\left(r\right)}}\;.$$
Here, as above, ν(k) is the $k^{\mathrm{th}}$ integer ≢0modq.

In Appendix C (see (Equation 61)), it is shown that such a sum can be written as follows
λβ=α1+α2∑u=−∞∞e2πix(q−1)u(qu)2+2Re∑t=1q−11−tq−1××∑u=−∞∞e2πix((q−1)u+t)α3(uq+t)2+α4cos(2πpt/q)(uq+t)2+α5sin(2πpr/q)(uq+t)3.

The first sum is calculated through the Bernoulli polynomial $B_{2}\left(x\right)$ (see (Equation 52)). The last sums are expressed through two functions
$$G\left(x,r\right)=\sum_{k=-\infty }^{\infty }\frac{e^{2\pi ix(kq+r)}}{{\left(kq+r\right)}^{2}}\;,\;F\left(x,r\right)=\sum_{k=-\infty }^{\infty }\frac{e^{2\pi ix(kq+r)}}{{\left(kq+r\right)}^{3}}\;.$$
The explicit expressions of these function can be obtained as follows.

Define one more function
$$g\left(x,r\right)=\sum_{k=-\infty }^{\infty }\frac{e^{2\pi ix(kq+r)}}{kq+r}\;.$$
By the differentiation over *x*, one has $G^{\prime }\left(x,r\right)=2\pi ig\left(x,r\right)$ and $F^{\prime }\left(x,r\right)=2\pi iG\left(x,r\right)$. As the differentiation of g(x,r) over *x* gives the sum of δ-function, it is plain that g(x,r) is the piece-wise constant function in interval [j/q,(j+1)/q]. Using (Equation 54), one gets
$$g\left(j/q+x,r\right)=\frac{\pi }{q\sin(\pi r/q)}e^{i\pi r(2j+1)/q}\;.$$
Correspondingly, function G(x,r) is a piece-wise linear function in the same intervals
(47)$$G\left(j/q+x,r\right)=\frac{\pi^{2}}{q^{2}\sin^{2}\left(\pi r/q\right)}e^{2\pi irj/q}+\frac{2\pi^{2}ix}{q\sin(\pi r/q)}e^{i\pi r(2j+1)/q}\;.$$
In the same way, one proves that function F(x,r) is a piece-wise quadratic function
(48)$$\begin{array}{ccc} F(j/q+x,r) & = & \frac{\pi^{3}\cos\left(\pi r/q\right)}{q^{3}\sin^{3}\left(\pi r/q\right)}e^{2\pi irj/q}+\frac{2\pi^{3}ix}{q^{2}\sin^{2}\left(\pi r/q\right)}e^{2\pi irj/q}-\\ & - & \frac{2\pi^{3}x^{2}}{q\sin(\pi r/q)}e^{i\pi r(2j+1)/q}\;. \end{array}$$
In all these formulas, j=0,…,q−1 and 0≤x≤1/q.

Combining all terms together, one finds
λβ=α1+2π2α2q2B2x(q−1)+2Re∑t=1q−11−tq−1××eixt/qα3+α4cos(2πpt/q)Gx(q−1)q,t+α5sin(2πpt/q)Fx(q−1)q,t.
The main interest for the calculation of the form factor is the behaviour of the largest eigenvalues for *x* close to zero. Using (Equation 47) and (Equation 48), one concludes that
$$\lambda_{\beta }=C_{0}+C_{1}\frac{\beta }{N_{1}^{\left(r\right)}}\;,\;0\leq \beta \ll N_{1}^{\left(r\right)}\;.$$
Here
$$\begin{array}{ccc} C_{0} & = & \alpha_{1}+\frac{\pi^{2}\alpha_{2}}{3q^{2}}+\\ & + & \frac{2\pi^{2}}{q^{2}}\sum_{t=1}^{q-1}\left(1-\frac{t}{q-1}\right)\left(\frac{\alpha_{3}+\alpha_{4}\cos\left(2\pi pt/q\right)}{\sin^{2}\left(\pi t/q\right)}+\frac{\pi \alpha_{5}\sin\left(2\pi pr/q\right)\cos\left(\pi t/q\right)}{q\sin^{3}\left(\pi t/q\right)}\right) \end{array}$$
and
$$C_{1}=-\frac{2\pi^{2}\alpha_{2}\left(q-1\right)}{2q^{2}}-\frac{2\pi^{2}\left(q-1\right)}{q^{2}}\sum_{t=1}^{q-1}\left(1-\frac{t}{q-1}\right)\left(\alpha_{3}+\alpha_{4}\cos\left(2\pi pt/q\right)\right)\;.$$
The sum over residues is of the form
$$\sum_{t=1}^{q-1}\left(1-\frac{t}{q-1}\right)h\left(t\right)$$
and (as it is easy to check) in the considered case h(q−t)=h(t). Therefore
$$\sum_{t=1}^{q-1}t\;h\left(t\right)=\sum_{t=1}^{q-1}\left(q-t\right)\;h\left(q-t\right)=\sum_{t=1}^{q-1}\left(q-t\right)\;h\left(t\right)⟶\sum_{t=1}^{q-1}t\;h\left(t\right)=\frac{q}{2}\sum_{t=1}^{q-1}h\left(t\right)\;.$$
Consequently
$$\begin{array}{ccc} C_{0} & = & \alpha_{1}+\frac{\pi^{2}\alpha_{2}}{3q^{2}}+\\ & + & \frac{2\pi^{2}\left(q-2\right)}{q^{2}\left(q-1\right)}\sum_{t=1}^{q-1}\left(\frac{\alpha_{3}+\alpha_{4}\cos\left(2\pi pt/q\right)}{\sin^{2}\left(\pi t/q\right)}+\frac{\pi \alpha_{5}\sin\left(2\pi pr/q\right)\cos\left(\pi t/q\right)}{q\sin^{3}\left(\pi t/q\right)}\right) \end{array}$$
and
$$C_{1}=-\frac{2\pi^{2}\alpha_{2}\left(q-1\right)}{2q^{2}}-\frac{2\pi^{2}\left(q-2\right)}{q^{2}}\sum_{t=1}^{q-1}\left(\alpha_{3}+\alpha_{4}\cos\left(2\pi pt/q\right)\right)\;.$$
Using sums indicated in Appendix C and collecting all terms in the end, one finds that
$$C_{0}=1,\;C_{1}=-2.$$

This result signifies that the largest moduli eigenvalues of the transition matrix for the barrier billiard with rational ratio h/b=p/q are (i) independent of values of integers *p* and *q* and (ii) have the same asymptotic expression as in (Equation 30) (taking into account that $N^{\left(r\right)}\approx 2N_{1}^{\left(r\right)}$)
$$\Lambda =\pm \left(1-\frac{\beta }{N_{1}^{\left(r\right)}}+O\left(\frac{1}{N_{1}^{\left(r\right)}}\right)\right)\approx \pm \left(1-\frac{2\beta }{N^{\left(r\right)}}\right)\;.$$
As it has been explained above, it implies that (i) the form factor for barrier billiards is different for odd and even *n* and (ii) the spectral compressibility is exactly equal 1/2 for all positions of the barrier
$$\chi \equiv K\left(0\right)=\frac{1}{2}\;.$$

The numerical calculations exemplified in Figure 6 confirm well these results.

## 7. Summary

It is demonstrated that the method of calculation of the level of compressibility proposed by G. Tanner in [18] for chaotic systems can successfully be applied for intermediate statistics models. The criterium discussed in [18] states that if the difference between the dominant eigenvalue of the transition matrix Λ=1 and the second in magnitude eigenvalue is big enough, then only the dominant eigenvalues contributes to the form factor, and one obtains the usual value of the form factor corresponding to standard random matrix ensembles. Notably, the level compressibility is zero.

For models considered in the paper, no individual transition matrix eigenvalues dominate, and one has to sum over many of them with moduli close to 1. Two types of random unitary matrices were investigated. The first corresponding to a quantisation of an interval-exchange map [20] has been discussed in detail in [21,22,23,24]. In particular, the values of the level compressibility were derived. The application of the transition matrix approach for this case serves first of all to check the validity of Tanner’s method for intermediate statistics models. It appears that interval-exchange matrices lead to circulant transition matrices whose eigenvalues are explicitly known and all necessary sums are easily estimated. In the end, one obtains the same values of the level compressibility as obtained in [21,22,23,24] but with much simpler and transparent calculations. An example is of a special interest. It corresponds to interval-exchange matrices with an irrational value of a parameter (which strictly speaking describes not an interval-exchange map but only a parabolic one). Numerically, it has observed in [22] that in such case, spectral statistics is of usual random matrix type (GOE or GUE depending on a symmetry) as for chaotic systems, which looks strange as the Lyapunov exponent of any parabolic map is zero. The transition matrix approach clearly indicates that although there is no dominant eigenvalue as has been discussed in [18], all eigenvalues except Λ=1 for large matrix dimensions are so quickly oscillating that averaging over a small interval of the argument effectively removes their contributions, producing the standard random matrix result.

The main part of the paper contains the calculation of the level compressibility for random unitary matrices derived from the exact quantisation of barrier billiards in [7,8]. The importance of such matrices comes from the fact that they have the same spectral statistics as high-excited states of barrier billiards, which are the simplest examples of pseudo-integrable models for which very little is known analytically.

The barrier billiard transition matrices are more complicated than the ones for interval-exchange matrices. Their spectra contain forbidden zones, and their exact eigenvalues seems not to be accessible in closed form. Nevertheless, as the level compressibility requires the control only of largest moduli eigenvalues of the transition matrix, it is possible to find such eigenvalues for large matrix dimensions precisely. The main simplification comes from the fact that eigenvectors corresponding to largest moduli eigenvalues are slow oscillating functions. Therefore, quickly oscillating terms in matrix elements will give negligible contributions on these eigenvectors, and one can substitute instead of exact matrix elements their average over fast oscillations. The resulting matrices are simpler and permit finding their large moduli eigenvalues analytically. In the end, one proves that the level compressibility of barrier billiards for all positions and heights of the barrier is the same and equals 1/2. This result strongly indicates that spectral statistics of the *B*-matrices associated with barrier billiards is universal (i.e., independent on the barrier position) and well described by the semi-Poisson distribution. 

## Figures and Tables

**Figure 1 entropy-24-00795-f001:**
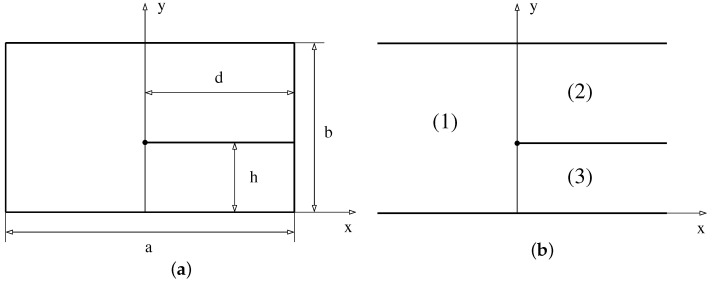
(**a**) Barrier billiard. (**b**) An infinite slab with a half-plane inside. Numbers indicate 3 possible channels.

**Figure 2 entropy-24-00795-f002:**
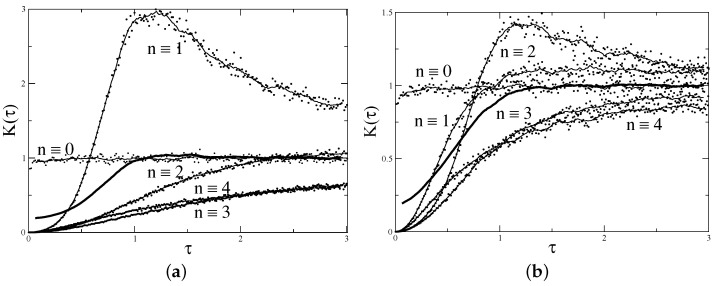
Form factor for the matrix (Equation 17) with α=1/5 and (**a**) N=399≡−1mod5 and (**b**) N=398≡−2mod5 averaged over 1000 realisations. Points are values of $K\left(n/N\right)$ for integers *n* with indicated residues modulo 5. Thin solid lines are a guide for the eye. Thick solid lines indicate the average over all 5 residues: $\frac{1}{5}\sum_{r=0}^{4}K\left((qt+r)/n\right)$.

**Figure 3 entropy-24-00795-f003:**
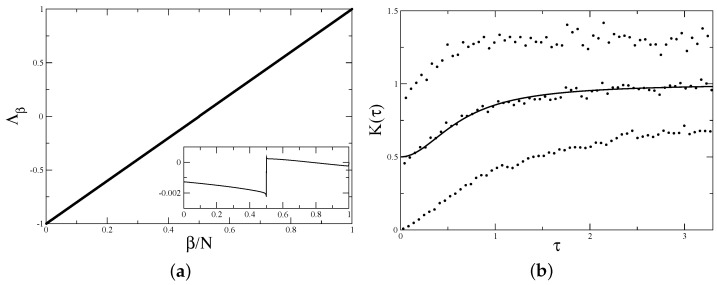
(**a**) Transition matrix spectrum for symmetric barrier billiard with h/b=1/2, b=π, k=1000.5, and N=1000. Insert shows the difference between true eigenvalues $\Lambda_{\beta }$ and the straight line 2β/N−1. (**b**) Form factor for symmetric barrier billiard with h/b=1/2, b=π, k=400.5, and N=400. Data are averaged over 1000 realisations of random phases. The upper dots are $K\left(2n/N\right)$, the lower dots are $K\left((2n-1)/N\right)$, and the middle dots correspond to $\left(K\left(2n/N\right)+K\left((2n-1)/N\right)\right)/2$. The solid line is the semi-Poisson prediction $K\left(\tau \right)=\left(2+\pi^{2}\tau^{2}\right)/\left(4+\pi^{2}\tau^{2}\right)$ [11].

**Figure 4 entropy-24-00795-f004:**
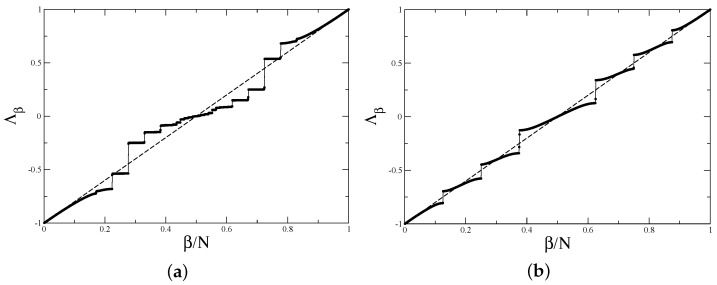
Spectra of the transition matrix for barrier billiard with (**a**) $h/b=1/\sqrt{5}$, b=π, k=500.5, and N=999 (cf. (Equation 4)) (**b**) h/b=2/5, b=π, k=650.5, and N=1040 (cf. (Equation 6)). The straight dashed line in the both figures is $\Lambda_{\beta }=2\beta /N-1$.

**Figure 5 entropy-24-00795-f005:**
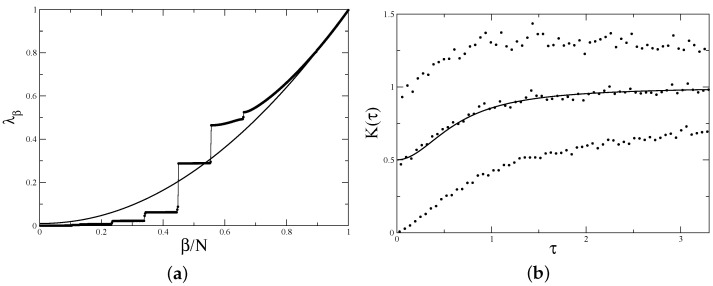
(**a**) Eigenvalues of the *P*-matrix (Equation 36), (Equation 37) for for $h/b=1/\sqrt{5}$ for the same parameters as in Figure 3a. Solid line indicates approximate expressions (Equation 39). (**b**) Form factor for $h/b=1/\sqrt{5}$, b=π, k=200.5, N=399 averaged over 1000 realisations. Other notations are as in Figure 3b.

**Figure 6 entropy-24-00795-f006:**
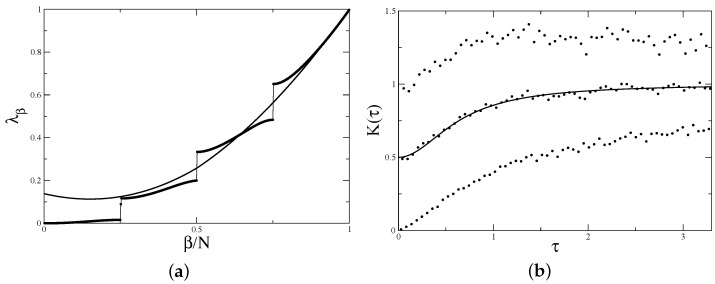
(**a**) Eigenvalues of the *P*-matrix (Equation 42), (Equation 43) for h/b=2/5 with $N_{1}=520$. Solid line is the spectrum (49) of the asymptotic matrix. (**b**) Form factor for h/b=2/5, b=π, k=250.5, and N=400 averaged over 1000 realisations. Other notations are as in Figure 3b.

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
