# Peer review of "Level Compressibility of Certain Random Unitary Matrices"

_entropy, 2022, doi:10.3390/e24060795_

Round 1

Reviewer 1 Report

The author calculates the level compressibility as defined from the number variance in a few unitary random matrix ensembles. What is of interest is

(a) the ensembles show intermediate statistics and (b) the techniques themselves. The author has written a clear and readable account of a difficult subject and made it accessible to many readers. The models themselves may be rather special, but I believe the generality of some of the ensembles such as a diagonal random matrix with a fixed unitary is very interesting. I expect that such ensembles will be of use in wider contexts. Indeed the semi Poisson statistics has also been found to hold in some instances of many-body quantum circuits for example:  https://journals.aps.org/prl/abstract/10.1103/PhysRevLett.112.240501

I recommend publication, the author may choose to include this and such references that may connect to non-traditional quantum chaos, or not!

Author Response

The first referee wrote: 

"The models themselves may be rather special, but I believe the generality of some of the ensembles such as a diagonal random matrix with a fixed unitary is very interesting. I expect that such ensembles will be of use in wider contexts. Indeed the semi Poisson statistics has also been found to hold in some instances of many-body quantum circuits for example:  https://journals.aps.org/prl/abstract/10.1103/PhysRevLett.112.240501"

\vspace{1cm}

The universality of the semi-Poisson statistics is not known at present. To stress its widespread  the following text has been added in the middle of page 5:  

"Despite the simplicity of the semi-Poisson distribution it has been observed (mostly numerically) in various  models ranging from certain pseudo-integrable models and quantum maps (see references in [7,8]) to the entanglement spectrum of two-bits random many body quantum circuits [12] ". 

Also a new reference suggested by the referee has  been added:

[12]  C. Chamon,  A. Hamma, and E. R. Mucciolo, \textit{Emergent irreversibility and entanglement spectrum statistics},
Phys. Rev. Lett. \textbf{112}, 240501, (2014). 

Reviewer 2 Report

The paper presents a detailed mathematical analysis of a quantity related to the spectral form factor. To me the quantity seems new and the results are original. The only comment or rather request to the author would be that he tried to put the new quantity into a physical context: why is it relevant to study this quantity rather than others well known ones? Would the compressibility be easier to measure for instance? What do I learn from it in general, in addition to what we know form the form factor or other quantities like the spectral rigidity or the like. 

Author Response

The second referee wrote: 

"The only comment or rather request to the author would be that he tried to put the new quantity into a physical context: why is it relevant to study this quantity rather than others well known ones? Would the compressibility be easier to measure for instance? What do I learn from it in general, in addition to what we know form the form factor or other quantities like the spectral rigidity or the like."
 \vspace{1cm}

I did not fully understand this remark. The compressibility is not a new quantity. It is just a long-range characteristic of the two-point correlation function and  can be extracted from any other quantities related with the two-point function  like the spectral rigidity. 

To explain better the situation the  sentences at page 5:  

"This paper is devoted to the calculation of another important characteristic of spectral
statistics, namely the level (or spectral) compressibility. This quantity is determined by the limiting behaviour of the variance of the number of levels inside a given interval." 

were changed as follows: 

"This paper is devoted to the calculation of another important characteristic of spectral
statistics, namely the level (or spectral) compressibility  which  is a long-range characteristic of the spectral two-point correlation function. In particular,  this quantity  determines  the limiting behaviour of the variance of the number of levels inside a given interval."